# Collaborative scheduling of dual-trolley quay cranes and AGVs via speed-control strategy

**Yao Tan[1], Fang Yan[2]\*, Xumei Zhang[1], Liu Yang[2], Feng Ma[3], Qi Wu[1]**

**1** School of Automobile and Traffic Engineering, Wuhan University of Science & Technology, Wuhan, Hubei, China, **2** Hubei Institute of Logistics Technology, Xiangyang, Hubei, China, **3** Key Laboratory of Metallurgical Equipment and Control Technology, Wuhan University of Science & Technology, Wuhan, Hubei, China

\* wlsyanfang@163.com

## Abstract

Conflicts may impede AGVs from reaching DQCs promptly during automated terminal operations. This challenge may lead to congestion on the transfer platform and diminish the operational efficiency of unmanned terminals. This research proposes a cooperative scheduling approach integrating a speed control strategy for DQCs and AGVs. Considering the capacity of transfer platforms, time windows for AGVs are established, and tasks are allocated accordingly. A speed control-based conflict resolution model is created with the dual objectives of minimizing energy consumption during travel and maximizing the fulfillment of time windows. The Dijkstra algorithm is employed to plan travel routes, anticipate potential conflicts during AGV operations, and assign priorities based on the satisfaction of task time windows. AGV speeds are dynamically adjusted to generate conflict-free scheduling plans that align with the operational times of the dual-trolley quay cranes. Experimental results demonstrate that the proposed speed-control strategy effectively resolves conflicts while consuming less energy than traditional stop-and-wait methods. Additionally, this strategy reduces the frequency of AGV starts and stops, ensures timely task completion, decreases quay crane waiting times, and enhances overall terminal operational efficiency.

## Introduction

As intermediate hubs connecting maritime and land transportation, container terminals play a pivotal role in global supply chains. Automated terminals have emerged as a prevailing developmental trend due to their notable advantages, including enhanced safety and reliability, superior operational efficiency, optimized space utilization, environmental sustainability, and reduced labor costs [1]. Automatic guided vehicles (AGVs) are a significant type of equipment in automated terminals, which

**Data availability statement:** All relevant data are within the paper and its Supporting Information files.

**Funding:** This research was supported by Chunhui Program Research Project of the Ministry of Education, (No. HZKY20220339), the Science and Technology Major Project of Hubei Province of China, (No. 42000024205T000000111), Hubei Provincial Major Science and Technology Project: Energy/Carbon Lean Accounting and Green-Smart Integration for Equipment Manufacturing Industry (No. 2023BCA006), The 14th Five-Year Plan Hubei Provincial Advantageous and Characteristic Discipline Clusters Project, (No. 2023B0405).

**Competing interests:** NO authors have competing interests.

can walk along a preset guidance path to complete a series of horizontal transport operations [2].

The Ministry of Transport's Guidelines on Accelerating the Development of Smart Ports and Smart Waterways (2023) emphasize the goal of comprehensively enhancing the digitalization of port infrastructure, intelligent management of production operations, and smart service delivery by 2027. Specific objectives include real-time monitoring of facility and equipment status, automated container transfer and stacking, intelligent planning and scheduling of transportation routes, and strengthened bidirectional information interaction to improve logistics efficiency. According to the latest statistics from the Ministry of Transport, China's total port container throughput reached 310.34 million TEUs in 2023, marking a year-on-year increase of 4.9%. Looking ahead, container throughput is projected to maintain stable growth on a high baseline, necessitating further improvements in terminal operational efficiency to meet escalating demands.

The dual-trolley quay crane (DQC) is a novel, high-efficiency container handling equipment. As illustrated in Fig 1, its primary components include a main trolley, a portal trolley, and a transfer platform. They collaboratively operate in a relay mode to load and unload containers. Uncertain events can trigger a series of chain reactions that can disrupt the entire automated terminal operation, causing deterioration of automated container terminal (ACT) efficiency [3]. A critical operational constraint arises when the transfer platform reaches its capacity limit, forcing the main trolley to halt operations. To prevent this, AGVs must arrive beneath the portal trolley within a predefined time window. Premature AGV arrivals lead to queuing delays, while late arrivals cause operational stoppages of the main trolley.

Resolving AGV conflict-induced stoppages and ensuring timely AGV arrivals to synchronize with the DQC's workflow. This improvement enhances collaborative efficiency between AGVs and dual-trolley DQCs, attracting extensive attention from researchers. This challenge is particularly critical in real-world logistics scenarios, where optimizing AGV scheduling to align with the DQC's time window remains a focal point of academic and industrial studies.

This study considers the transfer platform and its capacity constraints in dual-trolley quay cranes. Firstly, handling task time windows are determined based on the operational time of the crane trolleys, and these tasks are assigned to AGVs with travel paths planned using Dijkstra algorithm. Subsequently, multi-AGV conflicts are detected, and a dynamic priority strategy is applied to determine movement priorities. Finally, an improved speed control model is developed to regulate AGV speeds, resolving conflicts while simultaneously optimizing the satisfaction level of crane time windows and AGV energy consumption. This approach prevents situations where AGVs fail to reach designated trolley positions at the crane portal within scheduled timeframes due to conflicts.

The subsequent sections of this paper are organized as follows: Section 2 reviews the pertinent research on collaborative scheduling in container terminals. Section 3 details the modeling process in this study. Section 4 presents the case studies. Section 5 encapsulates the conclusions drawn from this research.

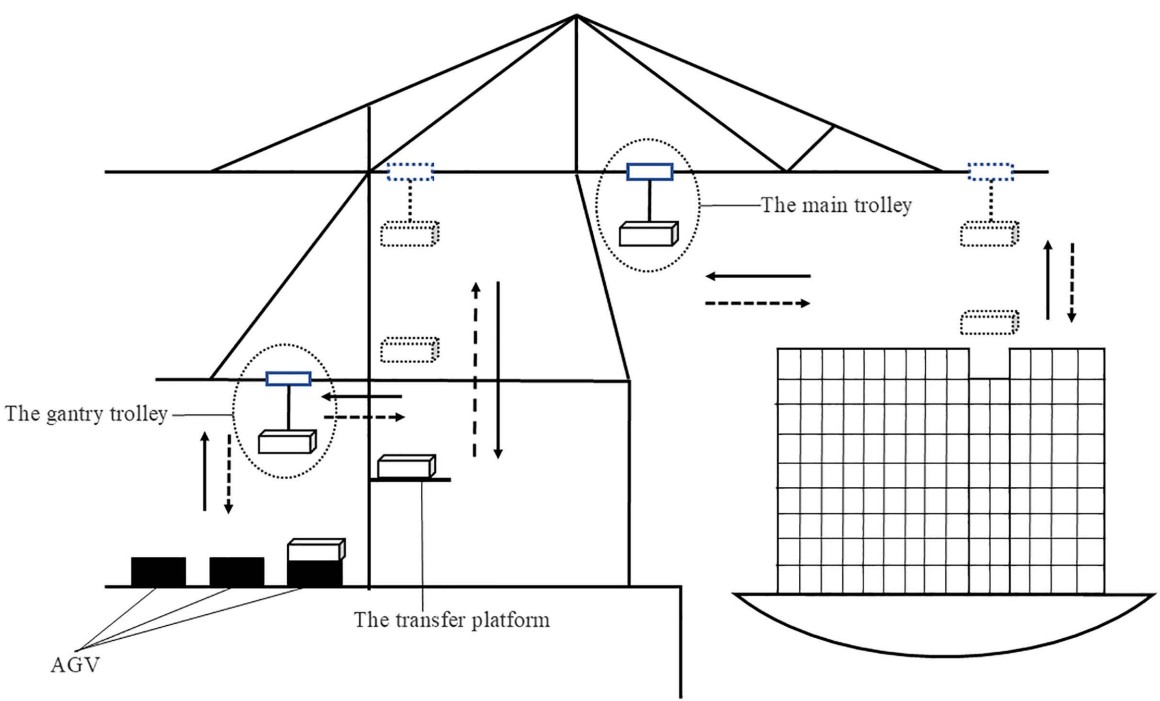

**Fig 1. Schematic diagram of dual-trolley QC.**

## Literature review

### Research on cooperative scheduling of DQCs and AGVs

In automated container terminals, various resources are highly interconnected and exhibit interdependent relationships, such that the scheduling of one resource inherently influences the scheduling of others [4]. In automated container terminals, the QC serves as the core equipment for container handling operations. Its operational efficiency directly determines vessel dwell times and exerts a profound impact on the overall terminal productivity [5]. The scheduling of QCs must not only fulfill loading/unloading requirements but also coordinate with other terminal equipment operations, particularly AGVs and yard cranes (YCs), to prevent resource conflicts and enhance operational fluidity [6]. As the critical transportation link between QCs and yard areas, the operational efficiency of AGVs is directly influenced by the loading/unloading pace of QCs [7]. Suppose QCs operate excessively fast while AGVs fail to deliver containers promptly. In that case, congestion may occur in terminal buffer areas or handshake zones, resulting in equipment waiting time and system bottlenecks [8]. Conversely, if QC operations are overly slow, AGVs may remain idle, reducing overall equipment utilization rates [9]. Therefore, optimizing collaborative scheduling between QCs and AGVs to minimize equipment idle time and improve terminal productivity has become a critical issue in current research [10].

In traditional scheduling optimization models, the transfer platform of QCs is typically treated as an independent constraint that governs task handover mechanisms between QCs and automated AGVs [9]. With advancements in automated terminals, scholars have proposed a novel optimization strategy that transforms the capacity constraints of QC transfer platforms into time window constraints for AGVs, aiming to achieve more efficient collaborative scheduling [11]. The core rationale lies in aligning QC operations with AGV transportation capabilities. By adjusting AGV task time windows to ensure timely arrival at transfer platforms, this approach reduces AGV waiting time and QC idle time [8]. Wei et al. investigated a scheduling optimization model under QC operational time window constraints, employing a mixed-integer linear

programming method to synchronize AGV operations precisely with QC loading/unloading activities [11]. This synchronization effectively reduces energy consumption and enhances overall terminal efficiency. Zhu et al. further developed a collaborative scheduling method based on buffer size and handshake zone positioning [8]. By optimizing AGV task allocation and execution sequences, their strategy enables AGVs to promptly receive or deliver containers upon QC operational completion, thereby minimizing efficiency losses caused by equipment waiting. This methodology addresses resource wastage stemming from inter-equipment coordination issues in conventional models, providing theoretical foundations for efficient collaboration among automated terminal equipment. Yue et al. proposed a cooperative scheduling optimization model for dual-trolley QCs and AGVs, establishing precise temporal synchronization between QCs and AGVs [9]. This ensures AGVs arrive beneath QCs within pre-defined time windows, significantly improving task handover efficiency. Their approach not only enhances QC operational continuity but also reduces AGV waiting time in transfer zones, thereby streamlining the entire workflow [9].

### Research on AGV conflict issues in container terminals

With the advancement of automation levels in port operations, AGVs have increasingly emerged as the primary horizontal transportation equipment within automated container terminals, and their efficient scheduling directly impacts terminal productivity [1,12,13]. However, path conflicts have emerged as a significant operational challenge due to the high-density deployment of AGVs and the complexity of terminal environments. To address this issue, scholarly efforts have predominantly focused on two key approaches: conflict-free path planning and obstacle-avoidance-based conflict resolution, further incorporating velocity control strategies to optimize conflict mitigation and enhance overall terminal productivity.

### Conflict-free path planning for AGVs in container terminals

Conflict-free path planning predefines AGV trajectories during task scheduling to ensure interference-free routes for all AGVs, thereby reducing potential conflicts and enhancing transportation stability and efficiency. Research efforts have also been devoted to solving the path conflict problem. Researchers have adapted the classic path planning methods (e.g., A* algorithm) to address path conflicts [14–16] and have investigated the performance of RL approaches to generate conflict-free paths [17]. Zhong et al. analyzed the impact of AGV strategy on path planning and constructed the mathematical model to prevent conflict and deadlock for multi-AGV scheduling [18]. Wang et al. investigated multi-AGV scheduling and path planning, proposing a branch-and-bound (B&B) algorithm to optimize task allocation and path selection, minimizing total operational time while guaranteeing collision avoidance [19]. Similarly, Xiong et al. developed a dynamic rolling scheduling model that leverages a spatiotemporal obstacle model to optimize AGV path planning for conflict minimization in static environments [20]. Hu et al. investigated the joint problem of scheduling and storage allocation, leveraging adjacency combinations and shortest-path principles to optimize AGV loading schedules in logistics distribution networks [21]. Additionally, Umar et al. introduced a priority-based genetic algorithm incorporating weight-mapped crossover (WMX) and insertion mutation strategies [22]. By encoding tasks with priority weights, this method enables AGVs to prioritize paths with the lowest conflict probabilities during the planning phase, enhancing both the feasibility and execution efficiency of path planning. The flow-paths were described as a square topology of AGV traffic network, and the control algorithm based on the chains of elementary reservations was proposed to prevent collision and deadlock by Małopolski [23].

### Dynamic adjustment to avoid AGV conflicts

Obstacle Avoidance-Based Conflict Resolution is primarily designed for dynamic path adjustments in fluctuating operational environments. By implementing local path modifications or task reallocations, this approach ensures AGVs maintain operational efficiency even in sudden conflict scenarios.

Speed control strategies improve system efficiency and optimize energy consumption by dynamically adjusting AGV velocities to prevent conflicts without altering predetermined paths. Fang et al. defined multi-AGV collision types and

proposed corresponding collision avoidance strategies, thereby enhancing the robustness of collaborative operations in multi-AGV systems [24]. Choe et al. developed an online preference learning algorithm capable of dynamically adapting scheduling strategies in response to evolving operational conditions [25]. Chen et al. developed a speed optimization model integrating autonomous truck platooning scheduling, enabling AGVs to reduce energy consumption through speed regulation while optimizing arrival time windows to minimize conflict probabilities [26]. Xing et al. advanced this concept through joint optimization of speed control and equipment scheduling, allowing AGVs to adapt velocities according to task urgency [27]. This approach effectively reduces waiting time at path intersections and decreases crane idle time, thereby enhancing operational efficiency. Yang et al. proposed a multi-objective optimization framework combining battery management with speed regulation, achieving concurrent optimization of path conflict mitigation and energy consumption management while maintaining operational requirements [28]. Adamo et al. introduced an integrated path-speed optimization methodology that coordinates AGV velocities to prevent simultaneous arrivals at intersections, thereby reducing congestion and conflict incidence [29]. Ji et al. further proposed a parking strategy to resolve AGV path conflicts, enhancing coordinated scheduling efficiency among quay cranes, yard cranes, and AGVs [30].

## Energy consumption optimization in terminal scheduling models

In recent years, green and low-carbon development in container terminals has driven energy-aware scheduling research. Zhang et al. propose that enhancing the efficiency of port operations must be accompanied by concurrent efforts to mitigate energy consumption [31]. Studies highlight that coordinated QC-AGV operations most significantly affect terminal energy consumption, with optimized scheduling reducing energy use [32]. Integrating energy factors into collaborative scheduling models to optimize energy use while ensuring efficiency remains a core focus. Liu et al. proposed a queuing theory-based QC allocation model to reduce AGV queuing and idling consumption [33]. Xin et al. minimized total energy consumption through joint QC-AGV task scheduling [34]. Zhao et al. optimized AQC-AGV task allocation under transfer platform capacity constraints to avoid congestion-induced energy waste [30]. Niu et al. reduced energy use by eliminating equipment conflicts and redundant movements [35], while Cai et al. enhanced AGV energy efficiency by optimizing path planning for productivity, obstacle avoidance, and minimal detours [36].

Recent studies on the coordinated scheduling of DQCs and AGVs in automated terminals have predominantly focused on optimizing the quantity configuration of DQCs and AGVs, as well as AGV path planning. Research that approaches coordination optimization from the perspectives of AGV fleet size and routing typically aims to minimize the maximum task completion time, subject to constraints such as the capacity of transfer platforms and AGV delivery times. Various algorithms are employed to derive detailed schedules for both DQC trolleys and individual AGVs. However, investigations that simultaneously guarantee scheduling coordination and detect AGV conflicts during actual delivery, while resolving these conflicts at low cost and energy expenditure, remain scarce.

## Model

### Problem description

In automated container terminals, the DQCs coordinate with AYCs through AGVs to accomplish ship-to-shore operations. During unloading operations, the DQC's main trolley transfers containers from the vessel to its transfer platform, while the gantry trolley subsequently loads them onto AGVs. Then AGVs transport containers to designated buffer zones at the yard interface, where YCs retrieve containers without queuing delays. This process is reversed during loading operations. The terminal layout integrating these components is illustrated in Fig 2.

The dual-trolley quay crane (QC) executes tasks according to a predefined operation sequence. However, when containers on the transfer platform reaches its maximum capacity, a blocking condition occurs. Neither the main trolley nor the gantry trolley can place additional containers onto the platform. This necessitates a waiting period until either the gantry trolley transfers unloading-task containers to AGVs, or the main trolley delivers loading-task containers to the vessel.

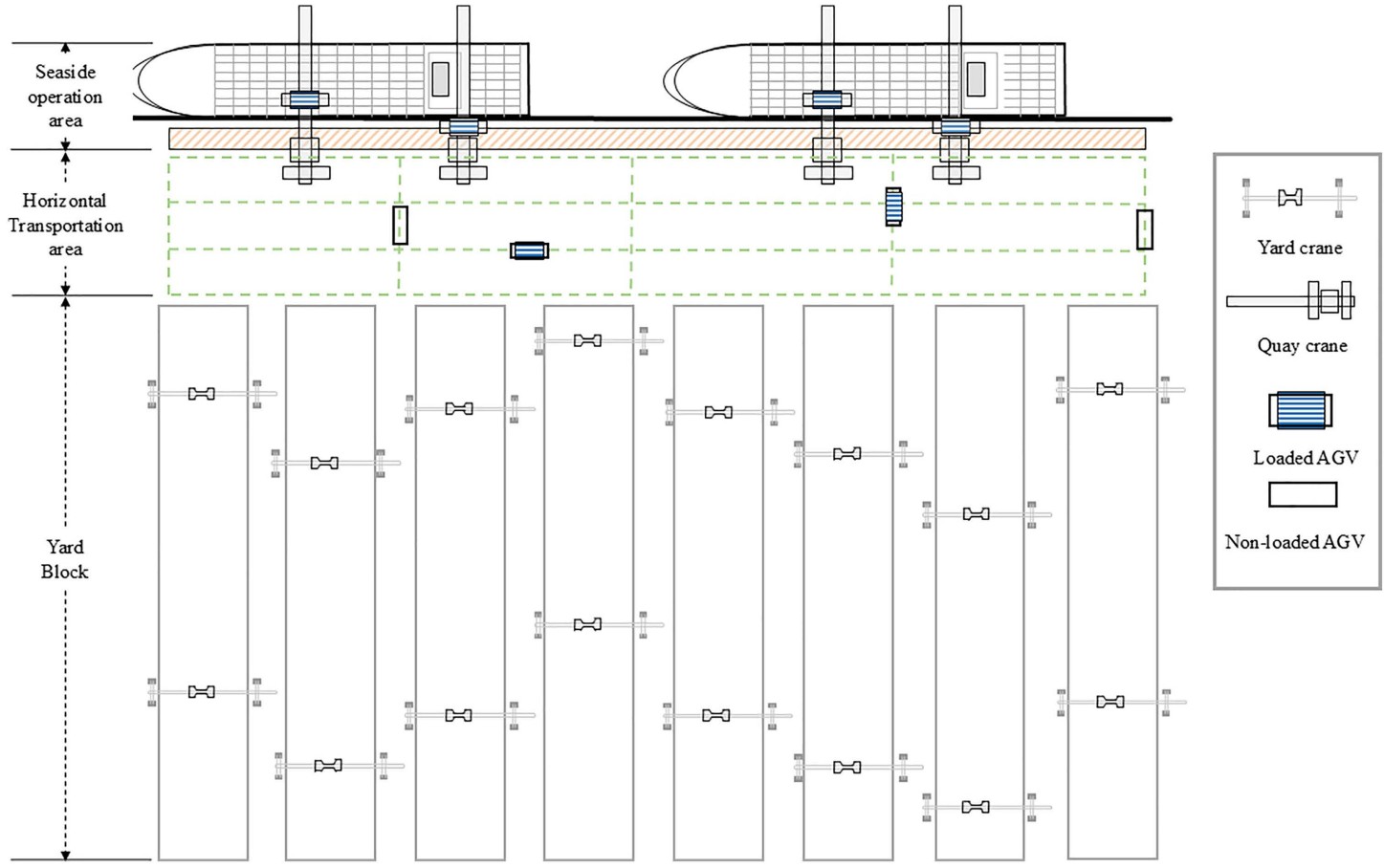

**Fig 2. Layout of dual-trolley QC automated terminal.**

In the collaborative scheduling process, the capacity constraints of the transfer platform are transformed into time window constraints for the gantry trolley. This conversion establishes task-specific operation time windows, while optimized AGV scheduling minimizes waiting times for both AGVs and the DQCs. Due to the structural constraints of terminal road network layouts and the limited route options for AGVs, traffic conflicts frequently occur during AGV operations, resulting in road congestion. This phenomenon leads to increased transportation duration and energy consumption, as well as reduced time window satisfaction rates for QC arrivals.

Based on converting the capacity constraints of the DQC transfer platform into soft time windows for AGV transportation tasks, this study rationally allocates tasks to individual AGVs. The Dijkstra algorithm has become an indispensable classic tool in the field of path optimization due to its core advantages, such as guaranteeing optimal solutions in graphs with non-negative weights, high computational efficiency, clear concepts, and ease of implementation. The Dijkstra algorithm is employed to determine optimal travel paths, while conflict identification and dynamic priority strategies are implemented to resolve operational conflicts. Subsequently, a speed control model is established with dual objective functions: minimizing AGV energy consumption and maximizing time window satisfaction. This model ultimately derives the gantry trolley operation time of DQC and generates a collaborative AGV scheduling scheme.

## Model assumption

To simplify problem modeling, we propose the following assumptions:

1. Each task point must be uniquely assigned to a single AGV, with duplicate operations at the same task point strictly prohibited. All tasks require full completion.

2. An AGV can only handle one container transportation task per operation cycle.

3. AGVs travel along unidirectional single-lane paths.

4. Operating AGVs must maintain minimum safe spacing from adjacent AGVs.

5. AGVs possess sufficient battery capacity to ensure continuous operation throughout horizontal transportation zone tasks.

6. All AGVs are restricted to unidirectional movement without reverse motion.

7. QCs are equipped with single-spreader systems, enabling only single-container operation mode.

## Definition of parameters and variables

The AGV transportation topology is represented by a weighted digraph $G = (N, W)$, $R_{a(n_i,n_j)} = (n_i, n_j)$ denoting the edge length of the $a$-th AGV from the node $i$ to the node $j$. The notations used in the following sections are shown in Table 1.

## Priority-based AGV speed control framework

Following predefined paths, the model predicts potential collision points along AGV paths. It then determines task priorities for conflicting AGVs and adjusts their velocities accordingly based on priority levels to resolve conflicts.

## Collision detection model

Three predominant conflict types exist in multi AGV path planning: head-on conflicts, catching-up conflicts, and node conflicts, shown in Fig 3.

Since the automated terminal in this study employs unidirectional guided paths, the analysis primarily focuses on node conflicts and catching-up conflicts.

1) Node Conflicts Detection Model:

$$C_{l,k} = N_l \cap N_k \tag{1}$$

$$TC = |t_{l,i} - t_{k,i}| < (L + L_s)/V_0 \tag{2}$$

Equation (1) represents the detection of spatial conflict points in the path. Equation (2) describes the temporal conflict point detection, where if one AGV arrives at a conflict point, the other AGV must have already passed through the point and traveled a safety distance L_s; otherwise, it is recorded as a node conflict.

2) Catching-up Conflicts Detection Model:

$$V_{l,ij} > V_{k,ij} \tag{3}$$

$$0 < L_{k,ij} - L_{l,ij} < L_s \tag{4}$$

**Table 1. Notations and explanations.**

| Notations | Explanation |
|---|---|
| $n$ | Total number of AGVs |
| $m$ | Total number of conflict nodes. |
| $M$ | The mass of the AGV. |
| $a$ | The index of the AGV. |
| $i$ | The index of the point. |
| $N_l$ | The set of nodes traversed by the $l$-th AGV. |
| $N_k$ | The set of nodes traversed by the $k$-th AGV |
| $C_{l.k}$ | The set of spatial conflict nodes between the $l$-th AGV and the $k$-th AGV. |
| $C_a$ | The set of spatial conflict nodes on the $a$-th AGV |
| $TC$ | The set of temporal conflicts occurring when AGVs arrive at conflict nodes. |
| $Q_1$ | Unit energy consumption cost. |
| $Q_2$ | Unit-time penalty cost. |
| $L$ | The length of AGV. |
| $L_s$ | Minimum safety distance between AGVs. |
| $V_0$ | Constant speed during normal AGV operation. |
| $V_1$ | Uniform speed during low-speed driving. |
| $V_2$ | Maximum safety-compliant driving speed. |
| $l_1$ | Deceleration distance — the distance an AGV travels while decelerating from $V_0$ to $V_1$. |
| $l_2$ | Acceleration distance — the distance an AGV travels while accelerating from $V_0$ to $V_2$. |
| $t_1$ | Deceleration duration of low-priority AGVs. |
| $t_2$ | Uniform-speed operation time post-deceleration for low-priority AGVs |
| $V_{a,ij}$ | Speed of the $a$-th AGV on path $(i,j)$. |
| $L_{a,ij}$ | Distance of the $a$-th AGV from node $i$ on path $(i,j)$. |
| $t_{ai}$ | Time at which the $a$-th AGV arrives at node $i$ under conflict-free conditions. |
| $t_{ae}$ | Actual task completion time for the $a$-th AGV. |
| $T_a$ | Optimal delivery time for the $a$-th AGV. |
| $T_{a,max}$ | Latest permissible delivery time for the $a$-th AGV. |

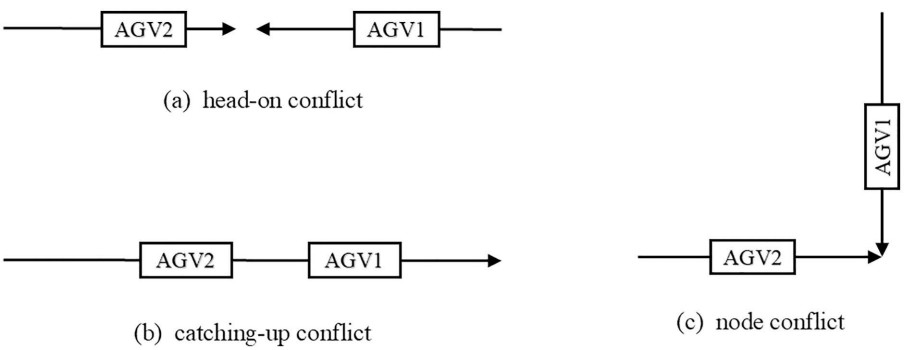

(a) head-on conflict

(b) catching-up conflict

(c) node conflict

**Fig 3. Three types of AGV path conflicts.**

Equation (3) indicates that the velocity of the $l$-th AGV on the path from node $i$ to node $j$ exceeds that of the $k$-th AGV at the same time instant. Equation (4) specifies that when the $k$-th AGV precedes the $l$-th AGV on the same path from node $i$ to node $j$ while their inter-vehicle distance falls below the safety threshold, and both conditions are satisfied simultaneously, this scenario is defined as a catching-up conflict.

## Dynamic priority assessment model

$$P_{li} = \alpha \cdot \frac{t_{l,e}}{T_l} + \beta \cdot \frac{t_{k,i}}{t_{l,i}} \tag{5}$$

$$P_{ki} = \alpha \cdot \frac{t_{k,e}}{T_k} + \beta \cdot \frac{t_{l,i}}{t_{k,i}} \tag{6}$$

Equation (5) defines the priority of the $l$-th AGV at the $i$-th conflict point, while Equation (6) determines the priority of the $k$-th AGV at the same conflict point. If $P_l < P_k$, the AGV on path $l$ is assigned a lower priority, whereas the AGV on path $k$ retains a higher priority. Here, $\alpha$ denotes the weighting coefficient for task execution time satisfaction, and $\beta$ represents the weighting coefficient for total travel time. Assigning higher priority to AGVs closer to the conflict point reduces the total travel time during conflicts, whereas prioritizing AGVs with later arrival times at their destinations enhances task execution time satisfaction.

## Velocity control strategy model

This study aims to minimize the energy consumption of AGVs and maximize time window satisfaction. After determining the transportation tasks and routes for each AGV, conflict points are identified, and AGV priorities are assigned based on the aforementioned conflict detection model and dynamic priority assessment model. High-priority AGVs accelerate to $V_2$ before reaching the conflict point to pass through it and subsequently decelerate to $V_0$. Low-priority AGVs decelerate to the reduced speed $V_1$ at a predetermined time before the conflict point, maintain speed $V_1$ for a period, and then reaccelerate to $V_0$ to pass through the conflict point. By the time the low-priority AGV passes through the conflict point, the high-priority AGV has already cleared the area, and both AGVs restore their original speed simultaneously, maintaining a safe distance to resolve the conflict. The energy consumption, comprising constant-speed and acceleration phases [35], is formulated as $P = 0.5 \cdot M \cdot V_1 \cdot t_2 + M \cdot I_1 \cdot (V_0 - V_1)$. A trapezoidal function is adopted to characterize the relationship between soft time windows and satisfaction levels, where solutions within the time window are prioritized, while violations are permitted with varying penalty intensities. Here, $(EET, EEL)$ denotes the acceptable arrival time window, and $(ET, EL)$ represents the desired arrival time window, as illustrated in Fig 4.

Therefore, the objective function of the model is formulated as follows:

$$minT = \omega_1 \cdot [0.5 \cdot M \cdot V_1 \cdot t_2 + M \cdot I_1 \cdot (V_0 - V_1)] \cdot Q_1 + \omega_2 \cdot |t_{l,e} - T_l| \cdot Q_2 \tag{7}$$

The constraints of the model are mathematically expressed as:

$$t_{l,i} - \frac{2I_1 + V_1 t_2}{V_0} + 2t_1 + t_2 = t_{k,i} - \frac{I_2}{V_0} + 2\frac{V_2 - V_0}{a} + \frac{L + L_s - I_2}{V_2} \tag{8}$$

$$V_1 = V_0 - at_1 \tag{9}$$

$$I_1 = V_0 t_1 - \frac{1}{2}at_1^2 \tag{10}$$

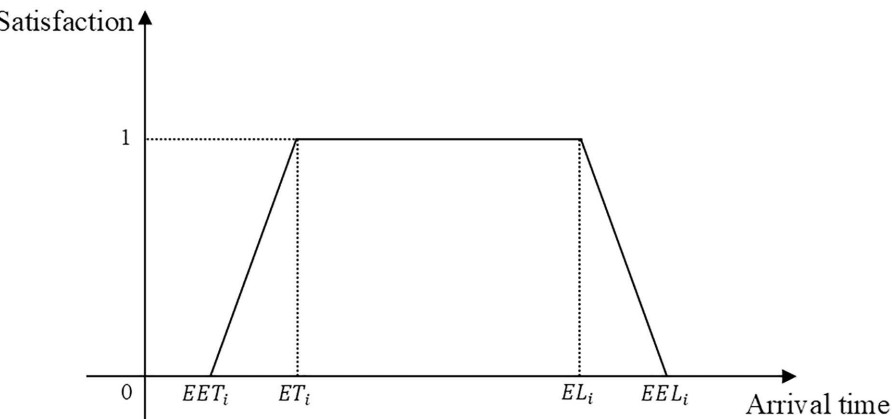

**Fig 4. Satisfaction-time profile.**

$$l_2 = \frac{V_2^2 - V_0^2}{2a} \tag{11}$$

$$t_{l,e} \leq T_{l,max} \tag{12}$$

$$t'_{k,j} = t_{k,j} + 2\frac{V_2 - V_0}{a} + \frac{L + L_s - l_2}{V_2} - \frac{l_2 + L + L_s}{V_0} \tag{13}$$

$$t'_{k,e} = t_{k,e} + 2\frac{V_2 - V_0}{a} + \frac{L + L_s - l_2}{V_2} - \frac{l_2 + L + L_s}{V_0} \tag{14}$$

$$t'_{l,j} = t_{l,j} + 2t_1 + t_2 - \frac{2l_1 + V_1 t_2}{V_0} \tag{15}$$

$$t'_{l,e} = t_{l,e} + 2t_1 + t_2 - \frac{2l_1 + V_1 t_2}{V_0} \tag{16}$$

Equation (8) states that the time for the low-priority AGV to reach the conflict point—after decelerating to velocity $V_1$, maintaining uniform motion at $V_1$, and then accelerating back to $V_0$—is equal to the time for the high-priority AGV, starting from a distance $l_2$ from the conflict point, to accelerate to $V_2$, traverse the conflict zone at $V_2$, and decelerate afterward. This synchronization ensures that the two AGVs maintain a safe distance during the interaction. Equation (9) defines the uniform reduced velocity $V_1$ adopted by the low-priority AGV. Equation (10) calculates the distance traveled by the AGV during deceleration to $V_1$, while Equation (11) computes the distance covered during acceleration from $V_1$ to $V_2$. Equation (12) constrains the actual arrival time of the low-priority AGV at its target point to not exceed the latest permissible delivery time. Equation (13) revises the arrival time of the high-priority AGV at subsequent node $j$ after implementing the speed control strategy. Equation (14) specifies the updated arrival time of the high-priority AGV at its destination post-strategy. Equation (15) updates the arrival time of the low-priority AGV at subsequent node $j$, and Equation (16) determines its modified arrival time at the target destination under the speed control strategy.

## Case study

Based on the terminal layout shown in Fig 2 as the prototype, the automated terminal planar layout is designed as a 4×9 topological diagram. This configuration primarily features unidirectional lanes arranged in an alternating pattern with counter-directional roadways. Four opposing longitudinal traffic routes are interwoven to simulate buffer zones in the terminal operation system. As illustrated in Fig 5, nodes 2 and 8 represent DQC locations, while nodes 29 and 35 designate the container yard areas. The system is equipped with four AGVs, initially positioned at nodes 2, 8, 29, and 35, respectively.

Most existing AGV conflict-avoidance studies that claim to employ speed control essentially revert to a "parking strategy," they impose a full stop at every potential collision point so that vehicles pass sequentially. To rigorously evaluate the proposed dynamic speed-control strategy, we conduct a systematic comparison with this parking strategy, focusing on the key performance indicators of energy consumption, makespan, and time-window satisfaction.

Table 2 presents the parameters used in the case, along with their descriptions and values. The terminal layout in this study and the parameter values listed in Table 2 are based on the corresponding data of Shanghai Yangshan Port [37].

### Time windows for dual-trolley quay cranes

In this study, the transfer platform capacity is configured as 2 containers. The main trolley requires 80 seconds per container move, while the gantry trolley operates at 60 seconds per move. Starting from time zero, the first container arrives at the transfer platform at 40 seconds, followed by the second at 80 seconds, with subsequent arrivals adhering to this sequential pattern. When the transfer platform reaches its full capacity of 2 containers, the main trolley operation is suspended. Given that DQC operational delays incur significant terminal efficiency losses, the AGV scheduling protocol mandates that AGVs must arrive punctually or ahead of schedule to eliminate main trolley waiting time. Consequently, containers must be promptly handed over to AGVs by the gantry trolley for delivery to yard blocks immediately after reaching the transfer platform. In the yard operation area, arrival time constraints are exempted due to the implementation of AGV companion devices. Based on these constraints, this research calculates optimal container task time windows to ensure coordinated scheduling between dual-trolley QCs and AGVs. The container handling schedule for the quay crane is detailed in Table 3.

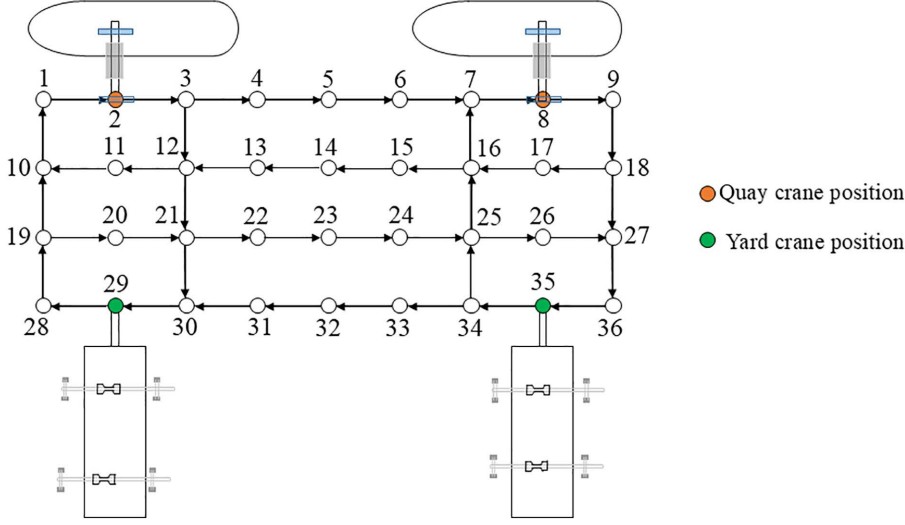

**Fig 5. Layout of AGV road network.**

**Table 2. Parameters, descriptions, and values in the case study.**

| Parameter | Description | Value |
|---|---|---|
| $L$ | Length of an AGV. | 6m |
| $l$ | Length of each grid path segment. | 20m |
| $V_0$ | Constant speed during normal AGV operation. | 4m/s |
| $V_2$ | Maximum safety-compliant driving speed. | 5m/s |
| $L_s$ | Minimum safety distance between AGVs. | 4m |
| $a$ | The AGV acceleration. | 1m/s2 |
| $v_t$ | The main trolley's operational speed. | 80s/move |
| $v_g$ | The gantry trolley's operational speed. | 60s/move |
| $c$ | The maximum containers stored of the transfer platform. | 2 |
| $t_{YC}$ | The YC operation time. | 10s |

**Table 3. Timetable for quay crane container handling.**

| Time(s) | Container handling status |
|---|---|
| 0 | Begin |
| 40 | Container 1- Transfer platform |
| 70 | Transfer platform – AGV |
| 90 | AGV – Yard |
| 120 | Container 2- Transfer platform |
| 150 | Transfer platform – AGV |
| 170 | AGV – Yard |
| 200 | Container 3- Transfer platform |
| 230 | Transfer platform – AGV |
| 250 | AGV – Yard |
| 280 | Container 4- Transfer platform |
| 310 | Transfer platform – AGV |
| 330 | AGV – Yard |
| … | … |

Under the constraint of preventing waiting time for the main trolley, preliminary calculations of AGV arrival times at the gantry trolley were conducted based on container unloading timelines from vessels to the transfer platform [38]. Starting from the moment the first container is transferred to an AGV (defined as time zero), the optimal AGV arrival times at QC2 and QC8 to complete four container-handling tasks were determined as 20s, 100s, 180s, and 260s, respectively. Consequently, the desired time windows for these tasks were set as (25, 35), (105, 115), (185, 195), and (265, 275), while the acceptable tolerance ranges were defined as (15, 45), (95, 125), (175, 205), and (255, 285).

## Task assignment and path planning

Based on the aforementioned task time windows, all tasks are allocated to the AGVs, and their traveling paths are determined using Dijkstra algorithm. The task allocation and path planning for the four AGVs are summarized in Table 4. Note that spatial conflict points are identified at nodes 7, 12, 16, 25, 27, and 30.

**Table 4. Task assignment and travel path scheduling scheme of AGVs.**

| AGV Index | Task Route | Travel Route | Expected Time Window |
|---|---|---|---|
| 1 | 2-35 | 2-3-4-5-6-7-8-9-18-27-36-35 | |
| | 35−8 | 35-34-25-16-7-8 | (105,115) |
| | 8-29 | 8-9-18-17-16-15-14-13-12-21-30-29 | |
| | 29−2 | 29-28-19-10-1-2 | (265,275) |
| 2 | 8-29 | 8-9-18-27-36-35-34-33-32-31-30-29 | |
| | 29−2 | 29-28-19-10-1-2 | (105,115) |
| | 2-35 | 2-3-4-5-6-7-8-9-18-27-36-35 | |
| | 35−8 | 35-34-25-16-7-8 | (265,275) |
| 3 | 29−2 | 29-28-19-10-1-2 | (25,35) |
| | 2-35 | 2-3-12-21-22-23-24-25-26-27-36-35 | |
| | 35−8 | 35-34-25-16-7-8 | (185,195) |
| | 8-35 | 8-9-18-27-36-35 | |
| 4 | 35−8 | 35-34-25-16-7-8 | (25,35) |
| | 8-29 | 8-9-18-27-36-35-34-33-32-31-30-29 | |
| | 29−2 | 29-28-19-10-1-2 | (185,195) |
| | 2-29 | 2-3-12-21-30-29 | |

## Analysis of AGV speed scheduling scheme results

The AGVs speed-time profiles are illustrated in Fig 6. As shown in the figure, AGV1 and AGV2 each execute one deceleration maneuver to avoid conflicts, while AGV3 performs two acceleration maneuvers and AGV4 completes one accelerated passage, collectively ensuring collision-free navigation for all AGVs. The conflict resolution process is detailed as follows:

The First Conflict: AGV1 and AGV4 encounter a spatial conflict at Conflict Point 7. The priority hierarchy is determined as $P_{1,7}=1.014<P_{4,7}=1.18$, assigning AGV1 a lower priority. AGV1 decelerates for 1.37s, maintains the reduced speed for 3.8s, then resumes its initial velocity. Concurrently, AGV4 accelerates to resolve the conflict.

The Second Conflict: AGV1 and AGV3 clash at Conflict Point 16. With $P_{1,16}=1.000<P_{3,16}=1.023$, AGV1 retains lower priority. AGV4 proactively accelerates to eliminate the conflict, allowing AGV1 to proceed without further speed adjustment.

The Third Conflict: AGV2 and AGV3 conflict at Conflict Point 7. The priority evaluation $P_{2,7}=1.001<P_{3,7}=1.018$ designates AGV2 as lower-priority. AGV2 decelerates for 1.23s, sustains the adjusted speed for 2.98s, and reverts to its original velocity, while AGV3 accelerates to bypass the conflict.

The time windows for AGV occupancy at spatial conflict points, DQCs, and yard blocks under different scheduling strategies are illustrated in Fig 7 and Fig 8, with dashed lines indicating the desired time window ranges.

As illustrated in the figure, the four AGVs demonstrate comparable numbers of assigned tasks and total completion times, with their arrival instants at quay cranes all falling within acceptable time windows. Each AGV occupies the operational zone for 60 seconds without instances of AGV-QC waiting interdependencies, indicating rational coordination in task sequencing, quay crane scheduling, and resource allocation. However, under the parking-wait strategy, the temporal deviations of AGVs' last two arrivals from expected time windows are notably greater than those observed in the velocity control strategy, manifesting as arrival delays that may propagate to subsequent material handling tasks. Furthermore, post-conflict analyses reveal that the parking-wait strategy induces prolonged occupancy of conflict points due to complete stops requiring extended restart durations, thereby exacerbating congestion risks. In contrast, the proposed speed control strategy enables lower-priority AGVs to decelerate preemptively before conflict points and subsequently resume nominal

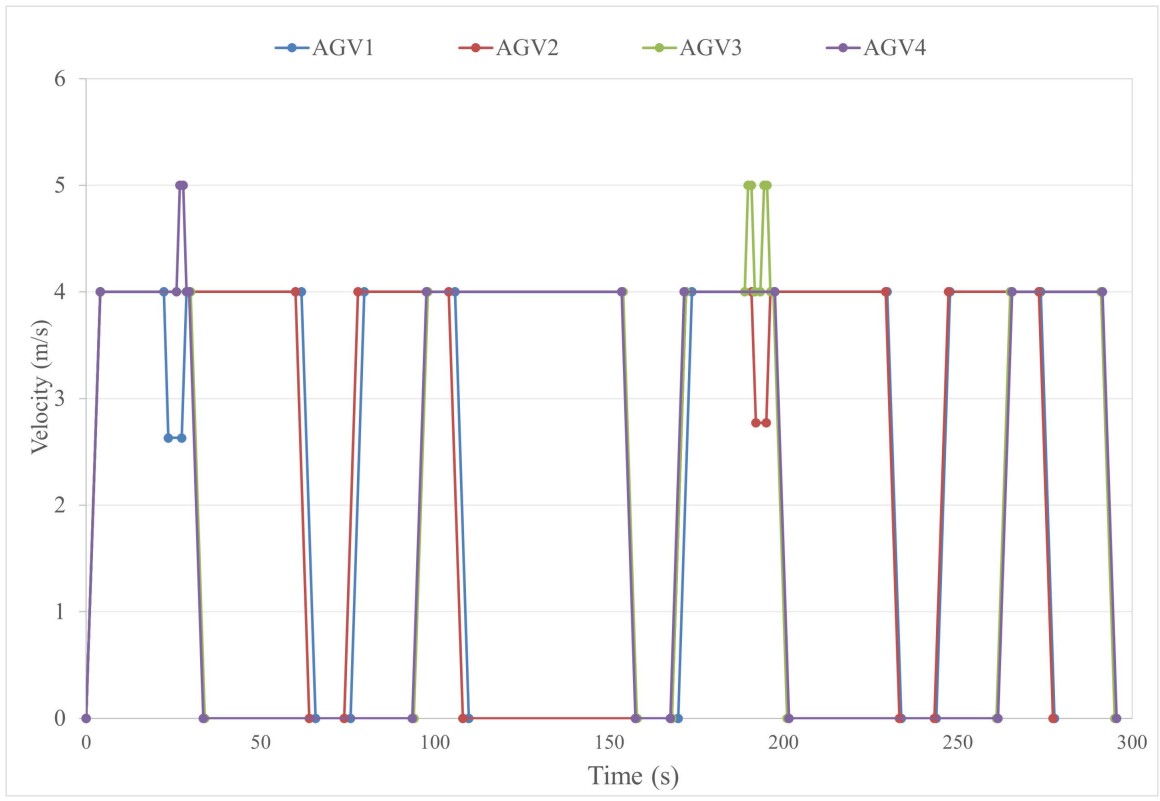

**Fig 6. Velocity-time chart of AGVs.**

transit speeds when traversing these zones. This approach eliminates extended occupation durations at conflict points while maintaining operational continuity, effectively mitigating potential congestion escalations.

The experimental results of different strategies are presented in Table 5. By comparing the proposed speed-control strategy with the traditional parking-wait strategy regarding total energy consumption, total travel time, and average time window satisfaction rate. The results indicate that the speed-control strategy and the parking-wait strategy exhibit comparable total travel time. However, the speed-control strategy demonstrates significantly superior performance in both total energy consumption and average time window satisfaction rate compared to the parking-wait approach. Therefore, when conflicts occur, the proposed speed-control strategy outperforms the parking-wait method, effectively reducing AGV energy consumption and addressing waiting issues at DQC interfaces.

## Comparative verification of two models

To validate the reliability of the speed-control strategy, a Monte Carlo simulation was conducted to compare it with the parking-waiting strategy. One thousand independent replications were performed at a 95% confidence level. Inter-arrival times of tasks were assumed to follow an exponential distribution, and the quay-crane service times were generated from a Gamma distribution, thereby capturing the operational dynamics observed in real container terminals.

The simulation results demonstrate that the speed-control strategy significantly outperforms the parking-and-waiting strategy across all performance metrics. As illustrated in Fig 9, the mean energy consumption under the speed-control strategy is 1701.65 J, 2.87% lower than the 1752.00 J recorded for the parking-and-waiting strategy.

**Fig 7. Occupancy diagram of speed-control strategy.**

In addition, as can be seen from Fig 10, the experimental average maximum completion time for the velocity control strategy is 295.5 seconds, which is 6.3 seconds shorter than that of the stop-and-wait strategy, and the time-window satisfaction rate increases markedly to 77.94% as shown in Fig 10 and Fig 11.

Moreover, the speed-control strategy achieves a superior constraint-satisfaction rate, improving the overall feasibility by 9.1 percentage points relative to the benchmark. The proposed strategy also exhibits stronger robustness and enhanced system stability.

## Conclusion

This study investigates the collaborative scheduling problem in automated terminals, considering transfer platform capacity constraints of DQCs. Task time windows are determined based on the operational timelines of DQCs. A multi-objective optimization model is formulated to minimize AGVs' total travel energy consumption and maximize time window satisfaction rates, incorporating conflict detection and resolution via speed-control strategy during AGV path assignments. Experimental results demonstrate that the proposed speed-control strategy effectively resolves conflicts by adjusting AGV speeds while ensuring energy-efficient arrivals at trolley interfaces within predefined time windows. This approach avoids

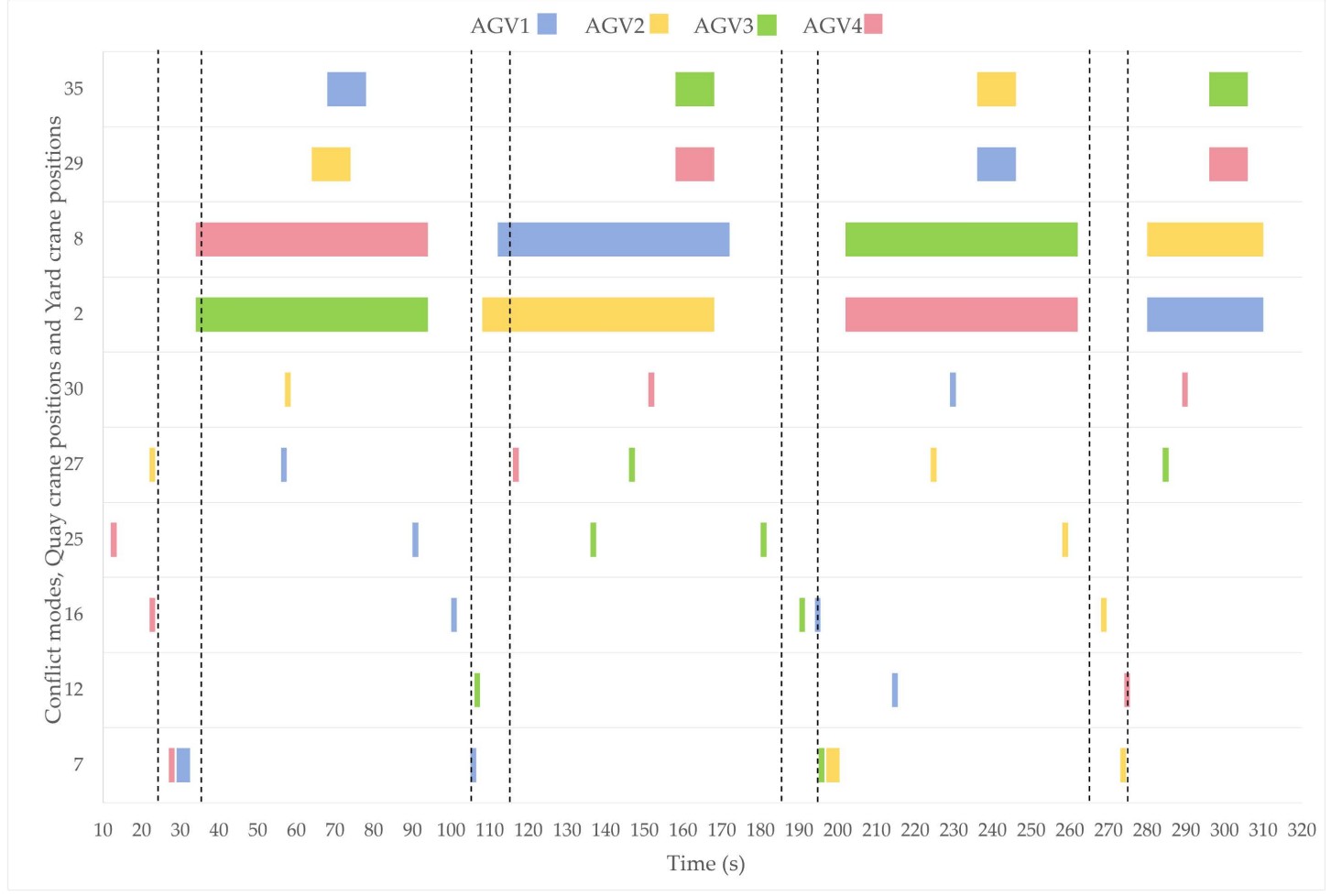

**Fig 8. Occupancy diagram of parking strategy.**

**Table 5. Comparison of different strategies.**

| | Proposed speed control strategy | Parking-wait strategy |
|---|---|---|
| **Energy consumption(J)** | 1701.65*mg | 1752*mg |
| **Travel time(s)** | 295.525 | 296 |
| **Average time window satisfaction rate** | 77.94% | 70% |

energy losses caused by the parking-wait strategy at conflict points and reduces congestion risks by minimizing prolonged occupancy of critical zones, thereby enhancing coordination between DQCs and AGVs.

By integrating a speed-control-based conflict-resolution mechanism, the proposed model simultaneously reduces energy consumption and improves time-window adherence, offering a practical scheduling solution for green and efficient operations in automated container terminals. In practical operations, AGV payload variations during delivery tasks

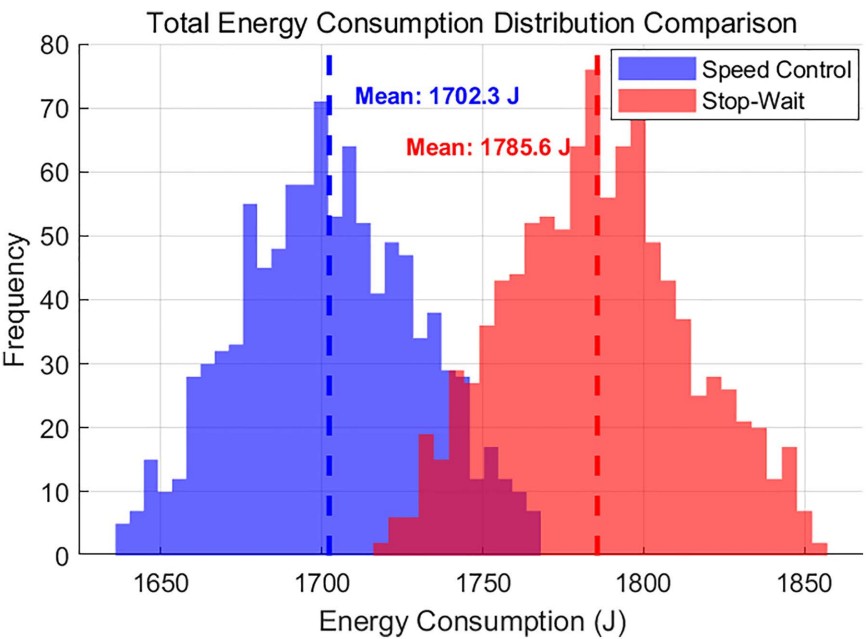

**Fig 9. Total energy consumption distribution comparison.**

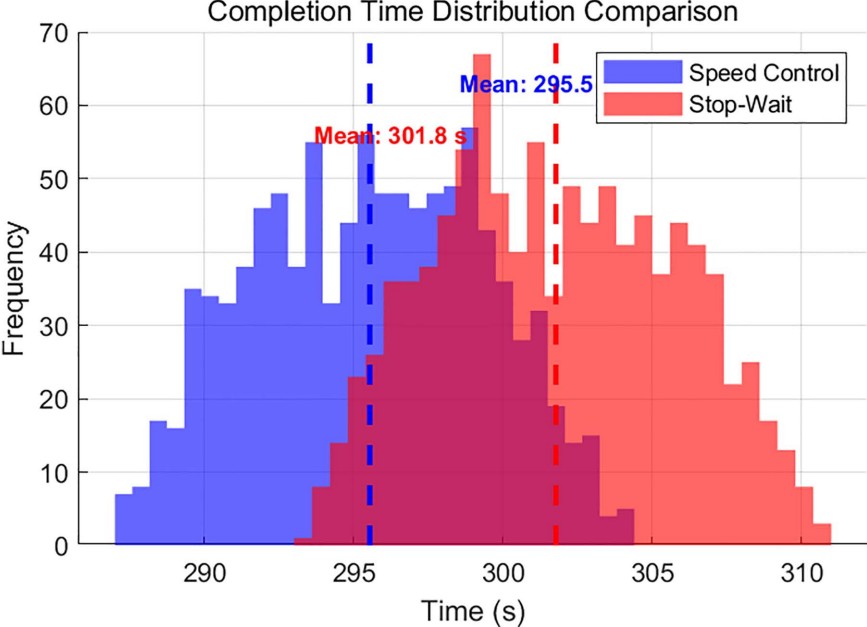

**Fig 10. Completion time distribution comparison.**

influence both energy consumption and travel speeds during conflict resolution. Additionally, AGV battery capacity constraints necessitate integrated charging scheduling and path planning. Future investigations should incorporate these factors to address real-world terminal scheduling requirements.

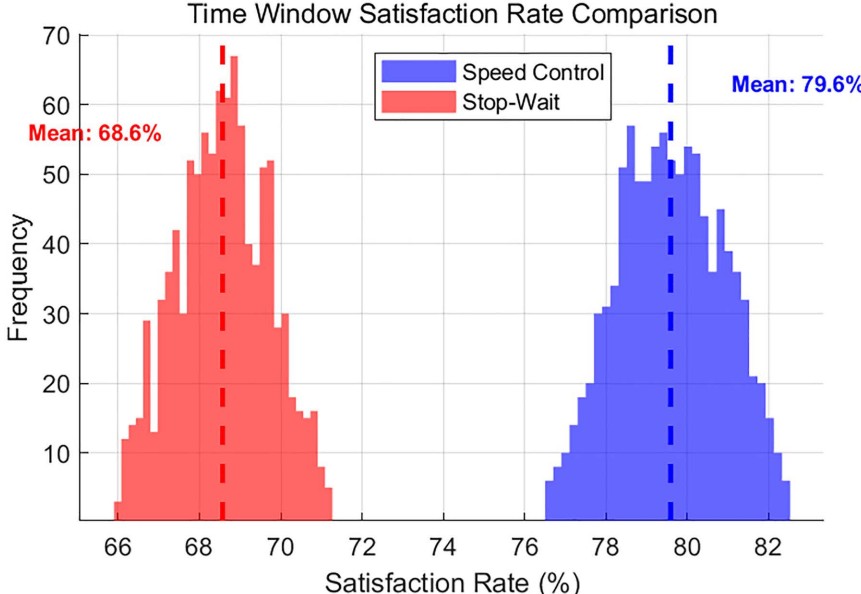

**Fig 11. Time window satisfaction rate comparison.**

## Supporting information

**S1 File. Models and data.**

(DOCX)

**S2 File. Original data in the study.**

(XLSX)

## Author contributions

**Conceptualization:** Yao Tan, Xumei Zhang.

**Funding acquisition:** Fang Yan, Xumei Zhang.

**Methodology:** Fang Yan, Liu Yang.

**Resources:** Fang Yan, Liu Yang, Feng Ma.

**Software:** Yao Tan, Qi Wu.

**Supervision:** Liu Yang.

**Writing – original draft:** Yao Tan, Qi Wu.

**Writing – review & editing:** Xumei Zhang, Feng Ma.

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
