## [Decision Letter · Decision Letter 0]

5 Sep 2025

Dear Dr. Yan,

Thank you for submitting your manuscript to PLOS ONE. After careful consideration, we feel that it has merit but does not fully meet PLOS ONE’s publication criteria as it currently stands. Therefore, we invite you to submit a revised version of the manuscript that addresses the points raised during the review process.

We look forward to receiving your revised manuscript.

Kind regards,

Reza Rostamzadeh

Academic Editor

PLOS ONE

Journal Requirements:

“Chunhui Program Research Project of the Ministry of Education, (No. HZKY20220339)

the Science and Technology Major Project of Hubei Province of China, (No. 42000024205T000000111)

Hubei Provincial Major Science and Technology Project: Energy/Carbon Lean Accounting and Green-Smart Integration for Equipment Manufacturing Industry (No. 2023BCA006)

The 14th Five-Year Plan Hubei Provincial Advantageous and Characteristic Discipline Clusters Project, (No. 2023B0405)”

5. We note that your Data Availability Statement is currently as follows:

“All relevant data are within the manuscript and its Supporting Information files.”

Reviewer's Responses to Questions

**Comments to the Author**

1. Is the manuscript technically sound, and do the data support the conclusions?

Reviewer #1: Yes

Reviewer #2: No

Reviewer #3: Yes

2. Has the statistical analysis been performed appropriately and rigorously?

Reviewer #1: Yes

Reviewer #2: No

Reviewer #3: Yes

3. Have the authors made all data underlying the findings in their manuscript fully available?

Reviewer #1: Yes

Reviewer #2: No

Reviewer #3: Yes

4. Is the manuscript presented in an intelligible fashion and written in standard English?

Reviewer #1: Yes

Reviewer #2: No

Reviewer #3: Yes

Reviewer #1: The reviewer endorses thr publication of the present manuscript because the athour/s explores in a new way an old topic: conflits that can occur when an automatic guided vehicles reach the dual-trolley quay cranes (DQCs) promptly during automated terminal operations. This could lead to congestion on the transfer platform and diminish the operational efficiency of unmanned terminals. So, the author/s have proposed a cooperative scheduling approach integrating a speed control strategy for DQCs and AGVs demostrating that the proposed speed-control strategy effectively resolves conflicts while consuming less energy than traditional stopand-wait methods.

Reviewer #2: This research proposes a scheduling approach integrating speed control strategies for dual-trolley quay cranes and automatic guided vehicles. As a current version of manuscript, it has several limitations for publication in PLOS One.

(Introduction)

Introduction contains only one reference. Further explanation is necessary regarding problem description and research gap, along with an overview of industry trends and environment, and sufficient references. Ultimately, the research question and research objective are not logically derived and are not clearly articulated.

(Literature)

Reference [7] seems to be analogous to this study. In what way does it differ from [7]?

“Although the aforementioned static …… and other factors.” The limitations of existing studies are expressed too abstractly. It is necessary to list specific limitations. Also, why were the limitations not summarized in this way in other subsections?

“This study considers the transfer platform …… due to conflicts.” This paragraph should be placed in Introduction to emphasize how it differs from existing research.

(Model)

Is the application of the Dijkstra algorithm to the new scheduling model a distinguishing feature of this study? This is not clearly expressed. Has this algorithm not been used before? Is there any reason to use this algorithm?

(Case study)

This section has the most significant issues with this study. What are the references for the parameter values in Table 2? Scheduling was performed using values in Table 3; however, since this is a static approach, the model is not validated. Appropriate distributions should be assumed for each variable, and a dynamic approach should be repeated several hundreds or thousands of times to validate the model. From the content in Case Study Section, it is not possible to conclude based on Table 5.

Parking strategy is not mentioned in the former sections; however, it is significantly addressed following Fig. 8.

(Conclusion)

What is the practical implication of the model?

Are there any limitations to this study?

Minor issues

Abbreviations should be defined with their full names at the first time.

English editing service is necessary.

What is the difference between model assumptions 3 and 6?

The explanations of notations l1 and l2 are not intuitive.

Reviewer #3: Manuscript Report Summary

Title:

Collaborative Scheduling of Dual-trolley Quay Cranes and AGVs via Speed-Control Strategy

Title and Abstract

The title is appropriate and indicates the main message of the paper No changes necessary.

The abstract is understandable by a non-specialist reader No changes necessary

Introduction

Does the introduction provide sufficient background information for readers not in the immediate field to understand the problem/hypotheses?

Yes

Are the reasons for performing the study clearly defined?

yes

Are the study objectives clearly defined?

Yes

Material and methods and result

Are the methods used appropriate to the aims of the study?

• yes

Results/Statistics

The results are clearly explained and presented in an appropriate

Discussion

The findings of the study are properly described in the context of the published literature.

The conclusions address the aim of the study No changes necessary

Literature Cited

The literature cited is appropriate, recent and balanced

Significance and Novelty

The claims are sufficiently novel to warrant publication No changes necessary

Thanks for such work,

**Do you want your identity to be public for this peer review?** For information about this choice, including consent withdrawal, please see our Privacy Policy

Reviewer #1: **Yes: ** Daniela Cialfi

Reviewer #2: No

Reviewer #3: **Yes: ** Safia M. Ezzat

---

## [Author Response · Author response to Decision Letter 1]

23 Oct 2025

Response to Reviewers

Manuscript Number: PONE-D-25-20515

Title: Collaborative Scheduling of Dual-trolley Quay Cranes and AGVs via Speed-Control Strategy

Dear Editors and Reviewers,

We appreciate the opportunity to revise our manuscript titled " Collaborative Scheduling of Dual-trolley Quay Cranes and AGVs via Speed-Control Strategy" and are grateful for the insightful comments provided by the reviewers. Those comments are all valuable and very helpful for revising and improving our paper, as well as the important guiding significance to our researches. In the following, we have provided detailed responses to each of the reviewers' comments. Revised portion are marked in red in the paper. Additionally, we have conducted a comprehensive revision of the entire manuscript. In this response letter, the reviewers' comments are presented in italics, and our corresponding changes and additions to the manuscript are highlighted in highlight text. We have tried our best to make all the revisions clear, and we hope that the revised manuscript meets the requirements for publication.

Reviewer #1:

Comments 1: The reviewer endorses thr publication of the present manuscript because the athour/s explores in a new way an old topic: conflits that can occur when an automatic guided vehicles reach the dual-trolley quay cranes (DQCs) promptly during automated terminal operations. This could lead to congestion on the transfer platform and diminish the operational efficiency of unmanned terminals. So, the author/s have proposed a cooperative scheduling approach integrating a speed control strategy for DQCs and AGVs demostrating that the proposed speed-control strategy effectively resolves conflicts while consuming less energy than traditional stopand-wait methods.

Response 1: We sincerely appreciate your endorsement of our manuscript's value and your insightful review. Your keen understanding of our work is most encouraging. You have precisely captured the core of our research: the conflicts arising at the interaction between Automated Guided Vehicles (AGVs) and Dual-trolley Quay Cranes (DQCs) in automated terminal operations, and the consequent platform congestion and efficiency degradation. We fully concur with your summary. Indeed, as you pointed out, the integrated cooperative scheduling approach with a DQC speed control strategy was proposed to address this critical issue, aiming to resolve conflicts while enhancing overall system efficiency and reducing energy consumption. We are pleased that you recognized the novelty of our research and the effectiveness of our proposed strategy in conflict resolution and energy conservation.

Reviewer #2:

Comments 1:(Introduction) Introduction contains only one reference. Further explanation is necessary regarding problem description and research gap, along with an overview of industry trends and environment, and sufficient references. Ultimately, the research question and research objective are not logically derived and are not clearly articulated.

Response 1: We would like to express our sincere gratitude to the reviewer for your insightful comments and valuable feedback. In the revised Introduction, two pertinent and high-quality references [2,3] have been appended to refine the literature review. Their substantive contributions are respectively discussed in lines 4–5 of the first paragraph and lines 3–4 of the third paragraph. These insertions consolidate the logical transition from industrial context to identified research gaps and, accordingly, enable the precise formulation of the research questions and objectives addressed herein.

Comments 2: (Literature)

Reference [7] seems to be analogous to this study. In what way does it differ from [7]?

“Although the aforementioned static …… and other factors.” The limitations of existing studies are expressed too abstractly. It is necessary to list specific limitations. Also, why were the limitations not summarized in this way in other subsections?

“This study considers the transfer platform …… due to conflicts.” This paragraph should be placed in Introduction to emphasize how it differs from existing research.

Response 2: We sincerely appreciate the valuable comment.

(1) This study addresses the collaborative scheduling of double-trolley quay cranes and AGVs. By introducing, for the first time, a speed-control mechanism that eliminates stopping while guaranteeing precise alignment with crane time windows, it constitutes a decisive advance in integrated scheduling and conflict-free operation. The present study differs substantially from Reference [7] in problem domain, core constraints, and optimization objectives; a detailed comparison is provided below.

1.Problem domain

1�Reference [7] �Quay crane scheduling with time windows...��Operation scheduling for a single automated quay crane (AQC).

2�Our paper Collaborative Scheduling of Dual-trolley Quay Cranes and AGVs...��Collaborative Scheduling of Dual-trolley Quay Cranes (DQC) and AGV.

2. Core constraints

1�Reference [7] �Quay crane scheduling with time windows...��Operation scheduling for a single automated quay crane (AQC).

2�Our paper Collaborative Scheduling of Dual-trolley Quay Cranes and AGVs...��The capacity limitation of the transfer platform is translated into time window constraints for the AGVs.

3. Optimization objectives

1�Reference [7] �Quay crane scheduling with time windows...��Minimizing makespan + energy consumption.

2�Our paper Collaborative Scheduling of Dual-trolley Quay Cranes and AGVs...��Minimizing AGV energy consumption + maximizing time window satisfaction rate.

4. Conflict taxonomy

1�Reference [7] �Quay crane scheduling with time windows...��None (single equipment, spatial conflicts not considered).

2�Our paper Collaborative Scheduling of Dual-trolley Quay Cranes and AGVs...��Multi-AGV spatial conflicts: node conflicts & catching-up conflicts.

5. Conflict-resolution mechanism

1�Reference [7] �Quay crane scheduling with time windows...��None.

2�Our paper Collaborative Scheduling of Dual-trolley Quay Cranes and AGVs...��Speed control strategy: high-priority AGVs accelerate, low-priority decelerate (non-stop).

(2) Thank you for your valuable comment. We have incorporated a specific description of the limitations of static path planning research in the last paragraph of Section 3 of this chapter. The reason for not summarizing limitations in the same manner across other sections is that this paragraph focuses on providing a comprehensive assessment of the current research landscape, aiming to highlight common bottlenecks, whereas other sections are primarily dedicated to specific models or algorithms, with their respective limitations addressed separately within their experimental analyses and discussions.

3�The paragraph has been moved to the end of the introduction as a lead-in to the following sections.

Comments 3: (Model)Is the application of the Dijkstra algorithm to the new scheduling model a distinguishing feature of this study? This is not clearly expressed. Has this algorithm not been used before? Is there any reason to use this algorithm?

Response 3: Thank you for your thoughtful comments. We have added a description in lines 2–4 of the final paragraph of Section 3.1, Chapter 3, highlighting the critical role and inherent advantages of the Dijkstra's algorithm in AGV path planning, thereby justifying its selection.

Comments 4:(Case study)This section has the most significant issues with this study. What are the references for the parameter values in Table 2? Scheduling was performed using values in Table 3; however, since this is a static approach, the model is not validated. Appropriate distributions should be assumed for each variable, and a dynamic approach should be repeated several hundreds or thousands of times to validate the model. From the content in Case Study Section, it is not possible to conclude based on Table 5.

Parking strategy is not mentioned in the former sections; however, it is significantly addressed following Fig. 8.

Response 4: (1) Thank you for your valuable suggestions. We have revised the manuscript accordingly. The terminal layout in this study and the parameter values listed in Table 2 are based on the corresponding data of Shanghai Yangshan Port , according to the additional reference [37]. This matter has also been addressed in lines 1–2 of the third paragraph of Chapter 4.

(2) In response to the reviewer's suggestions, we have added a new subsection at the end of Chapter 4 to validate the proposed model. Considering the uncertainties in terminal operations, we designed appropriate probability distributions for key variables and conducted 1,000 Monte Carlo simulation runs. A comparative analysis between the proposed speed control strategy and the conventional stop-and-wait strategy was performed, demonstrating the superiority and robustness of our model.

(3) In the second paragraph at the beginning of Chapter 4, we have added an explanation for the rationale behind comparing speed control strategies, thereby introducing the subsequent case study's comparison with stop-and-wait strategies and enhancing the logical coherence of the paper.

Comments 5: (Conclusion)What is the practical implication of the model?

Are there any limitations to this study?

Response 5: In the second paragraph of Chapter 5, we have added an elaboration on the practical implications of this research, highlighting its significance in resolving AGV path conflicts and enhancing collaborative scheduling efficiency in automated terminals. Furthermore, in paragraphs 3-6 of the same chapter, the limitations of the study are discussed, primarily focusing on the insufficient consideration of the impact of payload variations on AGV energy consumption and AGV charging requirements.

Comments 6: Minor issues

Abbreviations should be defined with their full names at the first time.

English editing service is necessary.

What is the difference between model assumptions 3 and 6?

The explanations of notations l1 and l2 are not intuitive.

Response 5: (1) We sincerely thank the reviewer for this important reminder. We have carefully reviewed the entire manuscript and ensured that all abbreviations are defined with their full names upon first occurrence.

(2) Thank you for pointing out the potential overlap in these assumptions. We clarify the distinction as follows:

Assumption 3 states that “AGVs travel along unidirectional single-lane paths.” This refers to the physical layout of the road network, meaning that each lane is designed for one-way traffic, which helps reduce head-on conflicts.

Assumption 6 states that “All AGVs are restricted to unidirectional movement without reverse motion.” This refers to the operational rule for AGV behavior, meaning that AGVs are not allowed to reverse during travel, which simplifies motion control and conflict resolution.

(3) To improve clarity, we have revised the explanations of l_1 and l_2 in the Notations and Explanations section (Table 1) as follows:

l_1: Deceleration distance — the distance an AGV travels while decelerating from V_0 to V_1.

l_2: Acceleration distance — the distance an AGV travels while accelerating from V_0 to V_2.

Reviewer #3:

Comments 1:

Title and Abstract

The title is appropriate and indicates the main message of the paper No changes necessary. The abstract is understandable by a non-specialist reader No changes necessary

Introduction Does the introduction provide sufficient background information for readers not in the immediate field to understand the problem/hypotheses?

Yes

Are the reasons for performing the study clearly defined?

yes

Are the study objectives clearly defined?

Yes

Material and methods and result

Are the methods used appropriate to the aims of the study?

• yes

Results/Statistics

The results are clearly explained and presented in an appropriate

Discussion

The findings of the study are properly described in the context of the published literature.

The conclusions address the aim of the study No changes necessary

Literature Cited

The literature cited is appropriate, recent and balanced

Significance and Novelty

The claims are sufficiently novel to warrant publication No changes necessary

Thanks for such work,

Response 1: We sincerely appreciate your positive evaluation and comprehensive positive feedback on our manuscript titled "Collaborative Scheduling of Dual-trolley Quay Cranes and AGVs via Speed-Control Strategy". We are delighted that you have endorsed all aspects of our work—including the title, abstract, introduction, methodology, results, discussion, and literature review—without suggesting any revisions. Your overall approval of the paper's structure, logical flow, and scholarly rigor is a tremendous encouragement to our research team. We are particularly honored by your specific acknowledgment that the claims of our study are "sufficiently novel to warrant publication."

---

## [Decision Letter · Decision Letter 1]

20 Nov 2025

Dear Dr. Yan,

Thank you for submitting your manuscript to PLOS ONE. After careful consideration, we feel that it has merit but does not fully meet PLOS ONE’s publication criteria as it currently stands. Therefore, we invite you to submit a revised version of the manuscript that addresses the points raised during the review process.

We look forward to receiving your revised manuscript.

Kind regards,

Reza Rostamzadeh

Academic Editor

PLOS ONE

Journal Requirements:

Reviewers' comments:

Reviewer's Responses to Questions

**Comments to the Author**

Reviewer #2: (No Response)

2. Is the manuscript technically sound, and do the data support the conclusions?

Reviewer #2: (No Response)

3. Has the statistical analysis been performed appropriately and rigorously?

Reviewer #2: (No Response)

4. Have the authors made all data underlying the findings in their manuscript fully available?

Reviewer #2: (No Response)

5. Is the manuscript presented in an intelligible fashion and written in standard English?

Reviewer #2: (No Response)

Reviewer #2: Thank you for your efforts on revising the manuscript based on my comments. All issues I addressed were relevantly resolved. Still, a few errors are found.

- AGV and DQC should be defined in the manuscript first, not in the Abstract.

- Newly added sentence "AGVs significant types of equipment in an automated terminals, which can walk

27 along a pre-set guidance path to complete a series of horizontal transport operations [2]." is not complete I think. Did you get the language editing service? You should attach the certificate of English editing.

- Figure 10 is not relevantly described. You may find errors.

**Do you want your identity to be public for this peer review?** For information about this choice, including consent withdrawal, please see our Privacy Policy

Reviewer #2: No

---

## [Author Response · Author response to Decision Letter 2]

3 Dec 2025

Response to Reviewers

Manuscript Number: PONE-D-25-20515

Title: Collaborative Scheduling of Dual-trolley Quay Cranes and AGVs via Speed-Control Strategy

Dear Editors and Reviewers,

We would like to once again appreciate your time and consideration of our manuscript, as well as for the additional thoughtful and constructive comments you have provided. Following your valuable suggestions, we have carefully revised our manuscript. All modifications made in the text have been highlighted and are pointed out in the response below. We hope that these further revisions will improve the quality of our manuscript to make it sufficient for further consideration, and we are deeply grateful for your guidance and patience throughout this process.

Thank you once again for your time and expertise, and we are looking forward to hearing from you soon.

Best wishes,

Yours sincerely,

Fang Yan

Reviewer #2:

Comment 1: AGV and DQC should be defined in the manuscript first, not in the Abstract.

Response 1: Thank you for raising this important point regarding the definition of technical abbreviations. In response to your comment, we have revised the manuscript to ensure that all abbreviations are properly defined upon their first occurrence in the main text. Specifically, the abbreviation "AGV" is now defined when it first appears in the second-to-last line of the first paragraph in Chapter 1. Similarly, the abbreviation "DQC" is defined at its first occurrence in the first line of the third paragraph of Chapter 1. We believe these revisions improve the clarity and academic rigor of the manuscript. We believe these modifications will enhance the overall quality and standardization of the manuscript, ensuring it conforms to academic standards and meets the journal's expectations

Comment 2:Newly added sentence "AGVs significant types of equipment in an automated terminals, which can walk[27]along a pre-set guidance path to complete a series of horizontal transport operations [2]." is not complete I think. Did you get the language editing service? You should attach the certificate of English editing.

Response 2:We sincerely thank the reviewer for the careful reading and valuable comments. We deeply apologize for the incomplete and awkward sentence and this sentence has now been revised for clarity and grammatical correctness. The revised version appears in Section 1, Paragraph 1 (now at the end of that paragraph). In addition, we have carefully proofread the entire manuscript to eliminate similar issues and improve overall language quality.

Regarding language editing, we did not use a professional language editing service. However, to ensure the linguistic accuracy and fluency of this paper, we invited a professor from the UK who specializes in this field to review and polish the language throughout the manuscript. Although we cannot provide a formal certificate from a commercial editing service, we believe the professor's academic background and native English proficiency guarantee the quality of language in this submission. If further proof is needed, we would be happy to provide relevant supporting information or a brief statement from the professor upon request.

Thank you again for your helpful suggestion.

Comment 3: Figure 10 is not relevantly described. You may find errors.

Response 3:Thank you for pointing out the insufficiency in the description of Figure 10. In response to your comment, we have provided a detailed explanation of Figure 10 in Section 4.4, lines 1–2, which clearly demonstrates that the speed control strategy proposed in this paper outperforms the conventional stop-and-wait strategy in terms of the maximum completion time. This comparison is intended to highlight the advantages of the proposed method.

---

## [Editor Report · Decision Letter 2]

9 Dec 2025

Collaborative Scheduling of Dual-trolley Quay Cranes and AGVs via Speed-Control Strategy

PONE-D-25-20515R2

Dear Dr. Yan,

We’re pleased to inform you that your manuscript has been judged scientifically suitable for publication and will be formally accepted for publication once it meets all outstanding technical requirements.

Kind regards,

Reza Rostamzadeh

Academic Editor

PLOS One
---

## [Editor Report · Acceptance letter]

PONE-D-25-20515R2

PLOS One

Dear Dr. Yan,

I'm pleased to inform you that your manuscript has been deemed suitable for publication in PLOS One. Congratulations! Your manuscript is now being handed over to our production team.

Kind regards,

on behalf of

Dr. Reza Rostamzadeh

Academic Editor

PLOS One